# Keep the fire burning: a survey study on the role of personal resources for work engagement and burnout in medical residents and specialists in the Netherlands

Lara Solms,[1,2] Annelies E M van Vianen,[1] Tim Theeboom,[3] Jessie Koen,[1] Anne P J de Pagter,[4,5] Matthijs de Hoog,[6] on behalf of the Challenge & Support Research Network

APJdP and MdH contributed equally.

This research has been presented at the Annual National Conference of the Netherlands Association for Medical Education (NVMO) in Egmond aan Zee, The Netherlands (16 November 2018).

For numbered affiliations see end of article.

**Correspondence to**
Dr Anne P J de Pagter;
p.depagter@erasmusmc.nl

## ABSTRACT

**Objectives** The high prevalence of burnout among medical residents and specialists raises concerns about the stressful demands in healthcare. This study investigated which job demands and job resources and personal resources are associated with work engagement and burnout and whether the effects of these demands and resources differ for medical residents and specialists.

**Design** In a survey study among residents and specialists, we assessed job demands, job resources, personal resources, work engagement and burnout symptoms using validated questionnaires (January to December 2017). Results were analysed using multivariate generalised linear model, ordinary least squares regression analyses and path analyses.

**Setting** Five academic and general hospitals in the Netherlands.

**Participants** A total number of 124 residents and 69 specialists participated in this study. Participants worked in the fields of pediatrics, internal medicine and neurology.

**Results** The associations of job and personal resources with burnout and work engagement differed for residents and specialists. Psychological capital was associated with burnout only for specialists ($b=-0.58$, $p<0.001$), whereas psychological flexibility was associated with burnout only for residents ($b=-0.31$, $p<0.001$). Colleague support ($b=0.49$, $p<0.001$) and self-compassion ($b=-0.33$, $p=0.004$) were associated with work engagement only for specialists.

**Conclusion** This study suggests that particularly personal resources safeguard the work engagement and lessen the risk of burnout of residents and specialists. Both residents and specialists benefit from psychological capital to maintain optimal functioning. In addition, residents benefit from psychological flexibility, while specialists benefit from colleague support. Personal resources seem important protective factors for physicians' work engagement and well-being. When promoting physician well-being, a one-size-fits-all approach might not be effective but, instead, interventions should be tailored to the specific needs of specialists and residents.

### Strengths and limitations of this study

► The participation of both residents and specialists from different specialties and the high response rate (78%) allow for a realistic display of the demands and resources present during different career stages.

► This is the first study to show that the personal resources psychological capital, psychological flexibility and self-compassion are important protective factors for residents and/or staff when fostering work engagement and combating burnout.

► This study reveals that different demands and resources relate to burnout and work engagement for specialists and residents, pointing to unique problems that require unique solutions.

► All data are based on self-reports, which limits drawing strong conclusions about objectively existing job characteristics and their associations with burnout and work engagement.

► Although theory and research point to causal relationships between demands and burnout and resources and work engagement, the study is limited by the fact that its cross-sectional design does not infer causality.

## INTRODUCTION

Healthcare professionals are well-known for their high work engagement, that is, their absorption by their work and dedication to patient care.[1 2] Unfortunately, such vigour and dedication can also have a flip side. Healthcare professionals work in a very complex system with long and irregular working hours and they need to deal with various high job demands such as time pressure, emotionally taxing patient interactions, work-family conflict and job insecurity. These high job demands cause alarming high rates of burnout symptoms such as emotional

**BMJ**

exhaustion and cynicism in healthcare compared with other professions.[3–7] Among physicians, medical residents seem to be especially at risk for burnout, with burnout symptoms often being reported by one out of four residents.[3 8–10] This is hardly surprising: in addition to the job demands that all physicians face, residents also face high educational demands, need to get used to clinical rotations and shifts, experience high responsibility yet limited autonomy and experience high uncertainty about their future career.[7 11] Additionally, residency marks a period that is characterised by stressful and demanding life events such as marriage and getting children.

Obviously, burnout has a strong negative impact on healthcare professionals and their employers. Most important, burnout symptoms are associated with decreased quality of care through delayed decision making, unprofessional work behaviours (ie, conflict) and suboptimal patient care (ie, not adequately discussing treatment options with patients or making medical errors).[7–9] Also, burnout is accompanied by long-term sickness leave and early retirement.[7] It is the highly critical work environment of healthcare professionals and the high-stake consequences of their functioning that call for a deeper understanding of the demands and resources that relate to the work engagement and well-being (ie, a lack of burnout) of these professionals. Similarly, it is important to understand which factors can help physicians to stay motivated in spite of the demands they face. An important result of motivation is work engagement, a positive work-related state of mind characterised by vigour, dedication and absorption.[1 12] Work engagement has been linked to employee performance across professions,[13 14] including healthcare.[15] Healthcare professionals who are engaged are also less likely to commit medical errors.[16] Consequently, both (a lack of) burnout and work engagement are important components for the optimal functioning of healthcare professionals.

Earlier research conceptualised burnout and work engagement as opposite poles of a continuum that are mutually exclusive.[17] Recent research[18] has evidenced that burnout and work engagement are (negatively) related, yet different constructs (ie, low burnout does not necessarily imply high engagement). Furthermore, these studies have shown that job characteristics that are associated with the prevalence of burnout are different from those that are associated with work engagement. Surprisingly, few of these studies were carried out in healthcare settings. Yet, knowledge of the job characteristics that relate to burnout and work engagement among healthcare professionals is essential to develop tailored interventions (eg, training, coaching) that support these professionals in their optimal functioning.[19]

### Theoretical Framework: The Job Demands—Resources Model

The Job Demands—Resources (JD-R) model[12] proposes that work engagement and well-being are promoted when (healthcare) professionals have job resources that help them to cope with high job demands and that bolster their motivation. Generally, the JD-R model differentiates between two universal types of characteristics that people find in their jobs: that is, job demands on the one hand and job resources on the other hand. Job demands refer to 'those physical, psychological, social or organisational aspects of the job that require sustained physical, and/or psychological effort and are therefore associated with physiological and/or psychological costs'.[12] Examples of job demands are workload, time pressure and emotional demands.[12] High job demands require professionals to spend sustained effort in order to meet perceived demands, which gradually drain resources and ultimately lead to depletion and exhaustion.[20] That being said, job demands are considered the prime factor leading to burnout.[12] Fortunately, professionals also have job resources that support them in coping with job demands. Such job resources, that are, 'physical, psychological, social or organisational aspects of the job that help to either achieve work goals, reduce job demands and the associated physiological and psychological costs, or stimulate personal growth, learning and development'[20] can comprise both situational/external and personal/internal resources.[21 22] Situational resources are, for example, colleague and supervisory support, and the amount of autonomy professionals have in their work.[18 23] However, resources are not exclusively found in the environment, but people can also create them for themselves.[24] Personal resources refer to individual psychological states such as an individual's psychological capital (including self-efficacy, hope, optimism and resilience), self-compassion (treating oneself with kindness when things go wrong) and psychological flexibility (the ability to choose behaviours that are in line with one's goals and values) that mirror people's perception to control and impact successfully on the environment.[25 26]

Apart from supporting employees in coping with job demands, situational and personal resources are also important in their own right as they are considered the prime factor leading to work engagement.[27] Meta-analytic studies have shown, for example, that colleague and supervisory support[13] and optimism and self-efficacy[28] were positively related to work engagement.

The current study builds on the JD-R model that considers burnout and work engagement as independent yet correlated constructs and job demands and (situational and personal) resources as the main predictors of burnout and work engagement, respectively. Although the relationships between demands and resources as antecedents and burnout and work engagement as outcomes are confirmed in numerous studies, research also shows that the strength of these relationships varies. This variation is likely due to the different professional samples and work contexts that were studied.[29–31] While residents and specialists work in the same occupational setting, they may face different job characteristics and, hence, different job demands and resources. As such, it is particularly relevant to examine whether the job demands, resources and outcomes, and the relationships among these variables,

differ between residents and specialists. Insight into this topic can inform stakeholders how to regulate workplace practices in order to foster physician well-being and work engagement. Importantly, such information can ensure the effectiveness of interventions because it allows us to tailor interventions to a specific situation or group and tap into personal and situational characteristics that can be changed at the individual and the group level, respectively. In the current study, we therefore investigate how job demands (workload, job insecurity, work-family conflict), situational resources (autonomy, supervisor support, colleague support) and personal resources (psychological capital, self-compassion, psychological flexibility) relate to work engagement and burnout among specialists and residents.

## METHODS
### Study population
From January to December 2017, we collected data from attending (specialists) and resident physicians at four academic hospitals and one general hospital in the Netherlands. The physicians were specialised or trained in pediatrics, internal medicine or neurology. Because this study is part of a larger programme offering individual coaching to physicians, the sample consists of physicians that signed up for the coaching programme or control participants. The choice of departments and hospitals that were invited for participation in this study was based on internal logistics. That is, because the coaching programme was only offered to physicians and residents from the pediatrics department, a relatively high number of participants in this sample are pediatricians. A comparison of the gender demographics of this sample with the broader population indicates that the sample of residents is representative of the hospital population. However, female specialists are over-represented in our sample. Limitations of generalisability will be discussed.

### Procedure
All physicians were invited by email to complete an online survey. Participation was voluntary; participants provided informed consent for participation in the study. We took measures to safeguard the anonymity and confidentiality of all participants.

### Measures
To capture the different components of the JD-R model, we included job demands, job resources, personal resources, burnout and work engagement in the survey, as well as demographics.

### Job demands
Job demands were assessed with three scales: workload, job insecurity and work-family conflict. *Workload* was assessed with four items from the Quantitative Workload Inventory[32] and two additional items (α=0.85). An example item measuring quantitative workload is: 'How

often does your job require you to work fast?' The two additional items were 'How often does your job require you to work overtime?' and 'How often do you experience emotional strain from your job?'. The items were scored on a five-point scale ranging from 1 ('never') to 5 ('always'). Higher scores indicate higher frequency, that is, higher workload.

*Job insecurity*, that is, 'the perceived threat of job loss and the worries related to that threat'[33] was measured with an adapted version of the Job Insecurity Scale.[34] The scale consisted of five items (α=0.83) including 'Chances are, that in the future I won't be able to find the job that I want' or 'I am feeling insecure about the future of my career'. The items were scored on a seven-point scale ranging from 1 ('not at all applicable') to 7 ('very applicable'). Higher scores indicate stronger applicability, that is, higher job insecurity.

*Work-Family Conflict* was measured with four items of the Work-Family Conflict Scale (α=0.87) measuring the extent to which 'the general demands of, time devoted to, and strain created by the work interfere with performing family-related responsibilities'.[35] An example item is: 'The demands of my work interfere with my home and family life'. The items were scored on a seven-point scale ranging from 1 ('not at all applicable') to 7 ('very applicable'). Higher scores indicate stronger agreement with the proposition, that is, higher work-family conflict.

### Job resources
Job resources encompassed autonomy, supervisor support and colleague support.

*Autonomy* was measured with nine items from the Work Design Questionnaire[36] (α=0.93) assessing perceived autonomy with regard to work scheduling and methods and decision-making. Example items include 'The job allows me to plan how I do my work', 'The job provides me with significant autonomy in making decisions' and 'The job allows me to make decisions about what methods I use to complete my work', respectively. The items were scored on a seven-point scale ranging from 1 ('totally disagree') to 7 ('totally agree'). Higher scores indicate stronger agreement with the proposition, that is, higher autonomy. *Supervisor support*, that is, the experienced psychological and work support from the supervisor, was assessed with six items from Vinokur *et al*[37] (α=0.95). Example items include 'My supervisor provides me with encouragement' or 'My supervisor says things that raise my self-confidence'. For residents, supervisory support measured the support received from the training supervisor, whereas for specialists supervisor support measured the support received from the head of the department. The items were scored on a seven-point scale ranging from 1 ('totally disagree') to 7 ('totally agree'). Higher scores indicate stronger agreement with the proposition, that is, higher supervisor support.

*Colleague support,* the experienced psychological and work support from colleagues, was assessed with the same six items as supervisor support (α=0.94), but the items

referred to colleagues instead of the supervisor. Also here, higher scores indicate stronger agreement with the proposition, that is, higher colleague support.

## Personal resources

We included three personal resources: psychological capital, self-compassion and psychological flexibility.

*Psychological capital* was measured with 12 items reflecting hope, optimism, resilience and self-efficacy from the validated Dutch version of the Psychological Capital Questionnaire[38 39] (α=0.88). The items include 'Right now I see myself as being pretty successful at work' (hope), 'I always look on the bright side of things regarding my job' (optimism), 'When I have a setback at work, I have trouble recovering from it, moving on (R)' (resilience) and 'When encountering difficult problems in my work, I know how to solve them' (self-efficacy). The items were scored on a seven-point scale ranging from 1 ('totally disagree') to 7 ('totally agree'). Higher scores indicate stronger agreement with the proposition, that is, higher psychological capital.

*Self-compassion*, entailing 'treating oneself with kindness, recognising one's shared humanity and being mindful when considering negative aspects of oneself',[40] was measured with six items from the Self-Compassion Scale[40] (α=0.72). Example items are: 'When I am going through a very hard time, I give myself the caring and tenderness I need' (self-kindness), 'I try to see my failings as part of the human condition' (common humanity) and 'When something painful happens I try to take a balanced view of the situation' (mindfulness). The items were scored on a five-point scale ranging from 1 ('rarely') to 5 ('almost always'). Higher scores indicate higher frequency, that is, higher self-compassion.

*Psychological flexibility,* that is, the ability to flexibly take appropriate action towards achieving goals and values, even in the presence of challenging or unwanted events[41] was measured with seven items of the Work Acceptance and Action Questionnaire[42] (α=0.81). Example items include 'I am able to work effectively in spite of any personal worries that I have' and 'I can work effectively, even when I doubt myself.' The items were scored on a five-point scale ranging from 1 ('rarely') to 5 ('almost always'). Higher scores indicate higher frequency, that is, higher psychological flexibility.

## Outcomes

The outcome variables included in this study were burnout symptoms and work engagement.

*Burnout* was measured with the Dutch version[43] of the Maslach Burnout Inventory—General Survey.[44] The instrument consists of three subscales measuring exhaustion, cynicism and professional efficacy. Because exhaustion and cynicism constitute the essence of the burnout syndrome,[45] we only measured these two components. *Exhaustion* was measured with five items (α=0.84). An example item is: 'Working all day is really a strain for me.' The items were scored on a seven-point scale ranging from 1 ('totally disagree') to 7 ('totally agree'). *Cynicism* was measured with four items (α=0.77). An example item is: 'I noticed that I have got too much distance from my work.' The items were scored on a seven-point scale ranging from 1 ('totally disagree') to 7 ('totally agree'). Higher scores indicate stronger agreement with the proposition, that is, higher exhaustion and cynicism, respectively.

*Work engagement*, including vigour, dedication and absorption at work, was measured with nine items from the Utrecht Work Engagement Scale[46] (α=0.90). Example items include: 'When I get up in the morning, I feel like going to work' (vigour), 'I am enthusiastic about my job' (dedication) and 'When I am working I forget everything around me' (absorption). The items were scored on a seven-point scale ranging from 1 ('never') to 7 ('always'). Higher scores indicate higher frequency, that is, higher work engagement.

## Statistical analysis
### Factor structure

To examine whether the items loaded on their respective scales, we performed separate confirmatory factor analyses (CFAs) for the scales representing job demands, job resources and personal resources, respectively. As each of these predictors consists of three scales, we compared a three-factor model to a one-factor model. We report the factor loadings and the commonly used model fit criteria, that is, the chi-square goodness-of-fit value, the chi-square divided by df (CMIN/DF), the comparative fit index (CFI), the root mean square error of approximation (RMSEA) and the standardised root mean squared residual (SRMR).

### Between-group variance

Because participants (n=192) can be considered as nested within (four) academic hospitals and (three) specialisations, we first assessed between-group variance within our data. A multilevel mixed-method analysis estimating a random intercept model was conducted to calculate between level-2 variance (The final Hessian matrix was not positive definite as the intercept variance was zero).

### Control variables

We explored the association between potential control variables (ie, age, gender, having children, job tenure, signed up for coaching) and the dependent variables by means of regression analyses for residents and specialists separately.

### Path analysis

The relationships between the independent (job demands, job resources and personal resources) and dependent variables (exhaustion, cynicism and work engagement) were examined with path analysis using IBM SPSS AMOS 25 (IBM SPSS, Chicago, Illinois, USA). In a first step, we modelled a latent variable termed burnout based on the observed variables exhaustion and cynicism. Modelling these two outcome variables on one latent variable was justified both theoretically[45 47] and statistically

(correlation of $r=0.58$, p<0.01 between exhaustion and cynicism).

A path model with independent variables and work engagement and burnout as dependent variables was tested using a covariance matrix as input and maximum likelihood estimation. This analysis adequately captures the nature of the associations between the independent and dependent variables and was therefore chosen over regular ordinary least squares regression analyses. Furthermore, this analysis allowed for a multigroup comparison, testing possible differences in model estimates between residents and specialists. Again, we report the commonly used model fit criteria as described earlier.

### Patient and public involvement

This study investigated factors associated with work engagement and burnout in medical specialists and residents. No patients or public representatives were involved in the study.

### RESULTS

In total, we invited a number of 247 physicians to take part in this survey of whom 75 physicians had signed up for a personal coaching programme that would start in a few months. A total number of n=193 physicians were included in this study after application of inclusion criteria (inclusion criteria: minimal response time >15 min, survey was filled out no later than 1 week after the first coaching session; survey progress ≥80%) (response rate=78%). The study population included 151 women (78.2%) and 42 men (21.8%). The mean age was 36.5 years ($SD$=8.5). One hundred and twenty-four residents (64.2%) and 69 medical specialists (35.8%) participated. Participants were working in the field of pediatrics (n=142; 73.6%), neurology (n=14; 7.3%) and internal medicine (n=37; 19.2%). See table 1 for a description of participants' characteristics. Internal consistency (Cronbach's alpha) was acceptable for all scales (see table 2).

### Factor structure

All items loaded on their respective scales. Factor loadings were on average 0.73, 0.81 and 0.60 for job demands, job resources and personal resources, with three items loading below 0.40 and a minimal loading of 0.28. These items were included because the scales were validated test instruments with overall high internal consistencies. The modification indices provided by the CFAs indicated that some items shared error variance. In order to improve the model fit, we allowed covariation of error variance between these items. Covariation was only allowed for items originating from the same scale. All three models provided adequate fit to the data with $\chi^2(86)=153.74$, p<0.001, $\chi^2/df=1.79$, CFI=0.95, RMSEA=0.06, SRMR=0.06; $\chi^2(180)=312.75$, p<0.001, $\chi^2/df=1.74$, CFI=0.97, RMSEA=0.06, SRMR=0.06 and $\chi^2(262)=435.96$, p<0.001, $\chi^2/df=1.66$, CFI=0.90, RMSEA=0.06, SRMR=0.07 for the three-factor models representing job demands,

**Table 1** Demographics of residents and specialists participating in a study on the role of personal resources for work engagement and burnout, 2017*

| Characteristics | Residents No (% of 124) | Specialists No (% of 69) |
|---|---|---|
| Gender | | |
| Female | 101 (81.5) | 50 (72.5) |
| Male | 23 (18.5) | 19 (27.5) |
| Age† | | |
| 20–30 years | 42 (33.9) | 1 (1.4) |
| 31–40 years | 81 (65.3) | 28 (40.6) |
| 41–50 years | 1 (0.8) | 22 (31.9) |
| 51–60 years | – | 15 (21.7) |
| 61 years and older | – | 3 (4.3) |
| Specialty | | |
| Pediatrics | 87 (70.2) | 55 (79.7) |
| Internal medicine | 27 (21.8) | 10 (14.5) |
| Neurology | 10 (8.1) | 4 (5.8) |
| Signed up for coaching‡ | | |
| Yes | 53 (42.7) | 36 (52.2) |
| No | 71 (57.3) | 33 (47.8) |
| Home situation | | |
| Children, one or more | 51 (41.1) | 48 (69.6) |
| No children | 73 (58.9) | 21 (30.4) |

*This study was conducted at four academic hospitals and one general hospital in The Netherlands. In this study, the authors investigated associations between job demands (workload, work-family conflict, job insecurity), job resources (autonomy, supervisor support, colleague support), personal resources (psychological capita, self-compassion, psychological flexibility) with work engagement and burnout.
†The residents taking part in this study were on average 31.9 years old (SD=3.0); the specialists were on average 44.9 years old (SD=8.7).
‡Participants were registered to participate in an institutional coaching programme or private coaching.

job resources and personal resources, respectively. Our results showed that the hypothesised three-factor model of these predictors provided a better fit to the data than a common factor model (eg, fit indices of the common factor model of job demands: $\chi^2(89)=864.96$, p<0.001, $\chi^2/df=9.72$, CFI=0.46, RMSEA=0.21, SRMR=0.19). The differences in the chi-square goodness-of-fit value between the three-factor and the common factor models were significant, all p's<0.001. These results allow us to conclude that the factor structure assumed in our path model is appropriate.

### Between-group variance

A multivariate generalised linear model analysis confirmed that hospitals and specialisations did not significantly differ on exhaustion, cynicism and work engagement. Therefore, it was not necessary to account

**Table 2** Correlations between the study variables for residents and specialists and internal consistencies of study variables in a study on the role of personal resources for work engagement and burnout, 2017†‡

| Study variables | 1 | 2 | 3 | 4 | 5 | 6 | 7 | 8 | 9 | 10 | 11 | 12 | 13 | 14 |
|---|---|---|---|---|---|---|---|---|---|---|---|---|---|---|
| 1. Signed up for coaching§ | – | 0.11 | 0.31* | −0.27* | −0.40** | 0.18 | 0.02 | 0.32** | 0.28* | 0.23 | 0.16 | −0.26* | −0.11 | 0.05 |
| 2. Job tenure | 0.13 | – | 0.02 | −0.22 | −0.10 | 0.14 | −0.05 | 0.11 | 0.19 | 0.21*** | 0.00 | 0.12 | 0.10 | 0.02 |
| 3. Workload | −0.09 | 0.08 | **0.85** | 0.18 | 0.46** | −0.14 | −0.00 | −0.16 | −0.14 | −0.16 | 0.06 | 0.33** | 0.27* | 0.02 |
| 4. Job insecurity | −0.16*** | 0.18* | 0.23* | **0.83** | 0.10 | −0.29* | −0.27* | −0.21*** | −0.53** | −0.42** | −0.29* | 0.29* | 0.38** | −0.41** |
| 5. Work–family conflict | −0.13 | 0.11 | 0.36** | 0.15 | **0.87** | −0.32** | −0.16 | −0.10 | −0.25* | −0.36** | 0.01 | 0.42** | 0.27** | −0.07 |
| 6. Autonomy | 0.13 | 0.13 | −0.29** | −0.18* | −0.11 | **0.93** | 0.21 | 0.29* | 0.55** | 0.37** | 0.18 | −0.35** | −0.28* | 0.38** |
| 7. Colleague support | 0.08 | −0.03 | −0.19* | −0.24** | −0.06 | 0.18* | **0.94** | 0.08 | 0.43** | 0.47** | 0.16 | −0.47** | −0.58** | 0.60** |
| 8. Supervisor support | −0.07 | −0.17 | 0.01 | −0.07 | 0.14 | 0.20* | 0.08 | **0.95** | 0.41** | 0.25* | 0.31** | −0.15 | −0.16 | 0.25* |
| 9. PsyCap | 0.13 | −0.13 | −0.22* | −0.38** | −0.21* | 0.33** | 0.26** | 0.19* | **0.88** | 0.68** | 0.37** | −0.50** | −0.60** | 0.58** |
| 10. Self-compassion | 0.13 | −0.13 | −0.32** | −0.41** | −0.33** | 0.15 | 0.22* | 0.09 | 0.51* | **0.72** | 0.25* | −0.33** | −0.50** | 0.32** |
| 11. Psych flexibility | 0.06 | 0.03 | −0.01 | −0.13 | −0.02 | 0.30** | 0.26** | 0.05 | 0.37* | 0.13 | **0.81** | 0.10 | −0.17 | 0.32** |
| 12. Exhaustion | −0.05 | 0.15*** | 0.35** | 0.22* | 0.51** | −0.04 | −0.23* | −0.07 | −0.32* | −0.45** | −0.26** | **0.84** | 0.67** | −0.50** |
| 13. Cynicism | 0.14 | 0.26** | 0.23** | 0.23* | 0.23* | −0.21* | −0.33** | −0.17 | −0.42** | −0.26** | −0.37** | 0.58** | **0.77** | −0.67** |
| 14. Work engagement | −0.05 | −0.04 | −0.12 | −0.15 | −0.14 | 0.31** | 0.24** | 0.20* | 0.54** | 0.29** | 0.38** | −0.40** | −0.62** | **0.90** |

Statistical significance of correlation coefficients is indicated by the following symbols: ***$p < 0.10$; *$p < 0.05$; **$p < 0.01$.

†Internal consistencies (Cronbach's alpha) are displayed on the diagonal in bold numbers.

‡The values below the diagonal refer to residents. The values above the diagonal refer to specialists.

§Participation (yes) was indicated with the number 1, participation (no) was indicated with the number 2.

PsyCap, psychological capital; psych flexibility, psychological flexibility.

**Table 3** Means and SD of study variables for residents and specialists in a study on the role of personal resources for work engagement and burnout, 2017

| Study variables | Residents Mean (SD)† | Specialists Mean (SD)‡ |
|---|---|---|
| Work demands | | |
| Workload*§ | 3.29 (0.68) | 3.52 (0.83) |
| Job insecurity** | 4.26 (1.17) | 3.03 (1.27) |
| Work-family conflict | 4.47 (1.14) | 4.56 (1.32) |
| Job resources | | |
| Autonomy** | 4.10 (0.99) | 5.19 (1.02) |
| Colleague support | 5.37 (0.85) | 5.38 (1.17) |
| Supervisor support | 4.66 (1.39) | 4.84 (1.49) |
| Personal resources | | |
| PsyCap*** | 4.91 (0.72) | 5.11 (0.75) |
| Self-compassion*** | 3.19 (0.63) | 3.36 (0.65) |
| Psych flexibility | 3.53 (0.61) | 3.62 (0.62) |
| Outcomes | | |
| Exhaustion | 2.55 (1.02) | 2.40 (1.18) |
| Cynicism | 2.24 (0.99) | 2.01 (1.07) |
| Work engagement* | 4.93 (0.77) | 5.21 (0.88) |

Differences in means between residents and specialists are indicated by the following significance values: ***p<0.10; *p<0.05; **p<0.01.
†A total number of 124 residents participated.
‡A total number of 69 specialists participated.
§Significance values were p=0.039 with equal variances assumed and p=0.052 with equal variances not assumed.
PsyCap, psychological capital; psych flexibility, psychological flexibility.

for group-level effects when estimating the relationships between the independent and dependent variables.

### Control variables
The results showed that only the control variables job tenure; $b=0.23$, $p=0.02$ (related to exhaustion for residents), $b=0.24$, $p=0.02$ (related to cynicism for residents) and signed up for coaching (response options were: 1=yes, 2=no); $b=-0.31$, $p=0.01$ (related to exhaustion for specialists) were related to exhaustion, cynicism or engagement. Therefore, and to save power, we only included job tenure and signed up for coaching as control variables in the further analyses.

### Descriptives and group differences
Table 3 describes means, SD and differences in study variables for residents and specialists, respectively.

Independent sample t-tests were performed to investigate mean-level differences in study variables comparing residents and specialists. Compared with specialists, residents reported significantly lower workload ($M=3.29$, SD=0.68 vs $M=3.52$, SD=0.83, $p<0.05$ (significance values were p=0.039 when equal variances were assumed and

p=0.052 when equal variances were not assumed)), lower autonomy ($M=4.10$, SD=0.99 vs $M=5.19$, SD=1.02, p<0.01) and lower work engagement ($M=4.93$, SD=0.77 vs $M=5.21$, SD=0.88, p<0.05). However, residents reported significantly higher job insecurity than specialists ($M=4.26$, SD=1.17 vs $M=3.03$, SD=1.27, p<0.01).

### Path analyses
#### Preliminary analyses
As suggested by Jöreskog and Sörbom,[48] we first specified an initial model based on our research question, and then adjusted the model according to the modification indices it produced, allowing covariation between all predictor variables, as well as covariation between job tenure and autonomy. Because both indicators of burnout highly correlated with engagement, we allowed covariation of error variance between exhaustion and cynicism with engagement. Testing the initial path model for specialists and residents separately revealed that the control variable signed up for coaching was not related to any of the two outcomes in both subsamples. We, therefore, removed this variable from the analysis and continued the analysis with only job tenure as control variable.

#### Model fit
The path analysis showed a satisfactory fit to the data, $\chi^2(51)=109.25$, p<0.001, $\chi^2/df=2.14$, CFI=0.96, RMSEA=0.06, SRMR=0.06. In order to improve the model fit, we removed the paths that were non-significant for both residents and specialists.[49]

We removed the paths from job insecurity and supervisor support to burnout. Further, we removed the paths from workload, job insecurity, work-family conflict, autonomy, supervisor support and job tenure to work engagement as they were not significant, partly despite significant zero-order correlations between these variables (see table 2). The model resulted in an improved fit of $\chi^2(75)=132.33$, p<0.001, $\chi^2/df=1.76$, CFI=0.96, RMSEA=0.05, SRMR=0.06. The tested model is presented in figure 1. The model explained 53.9% of the variance in burnout and 27.9% of variance in work engagement.

#### Relationships with burnout
The standardised path coefficients with burnout as outcome are presented in figure 1.

#### Job demands
Separate tests for residents and specialists suggested that there were no differences between both groups: workload was positively related to burnout for residents ($b=0.20$, p=0.011) and specialists ($b=0.22$, p=0.009). A multigroup comparison test confirmed that these relationships did not significantly differ, p>0.05. Furthermore, separate tests for both groups suggested that work-family conflict as a job demand differed between residents and specialists: it was positively related to burnout for residents ($b=0.33$, p<0.001) but not for specialists (p>0.05). However, these relationships did not significantly differ in a multigroup comparison test, p>0.05.

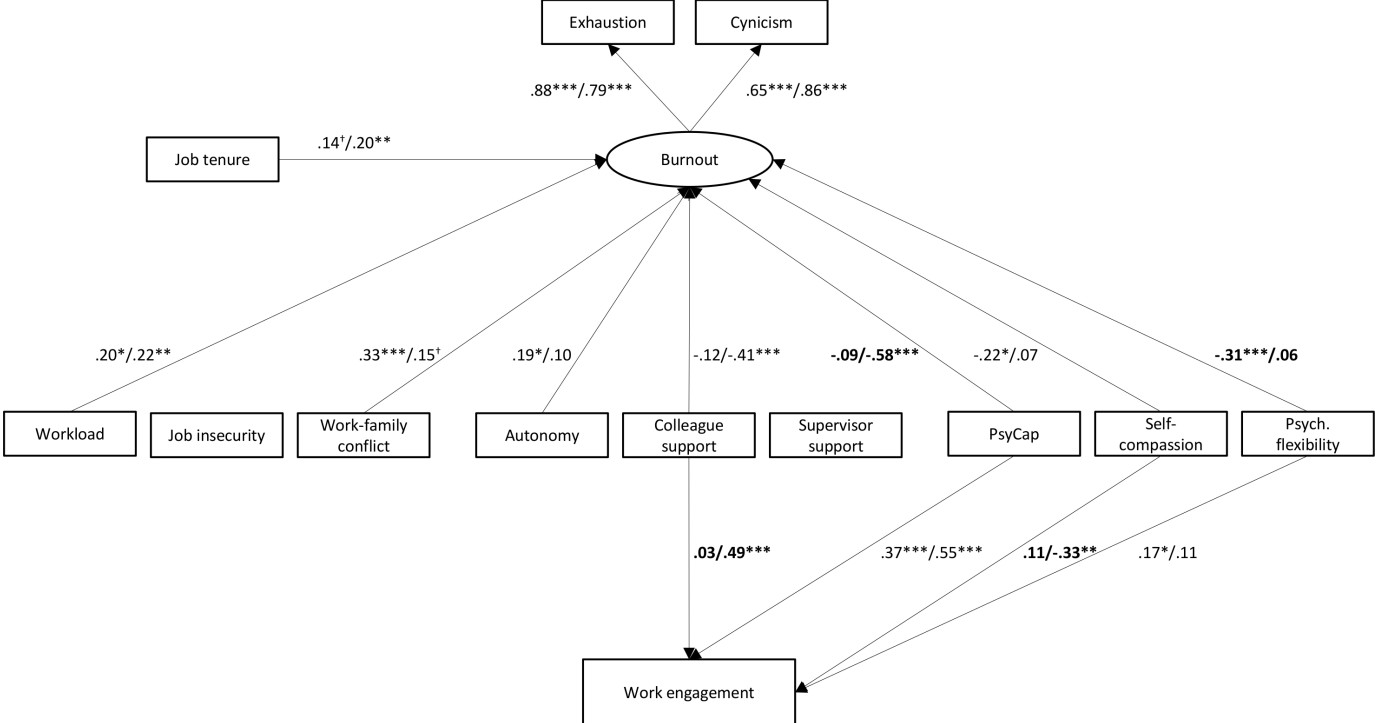

Abbreviation: PsyCap indicates psychological capital; psych. flexibility indicates psychological flexibility; the direction of the paths is indicated by an arrow symbol.
[a]Burnout as latent variable is based on the observed variables exhaustion and cynicism.
[b]The authors allowed covariation between all predictor variables, as well as between job tenure and autonomy. They also allowed covariation of error variance between exhaustion and cynicism with work engagement. For ease of reading, covariation is not depicted in the model.
[c]Only significant paths are displayed; the paths from job insecurity and supervisor support to burnout were not significant. Also, the paths from work-family conflict, autonomy, supervisor support, job insecurity and workload to work engagement were not significant.
[d]Standardized parameter estimates refer to residents (first) and specialists (second).
[e]Statistical significance of parameter estimates is indicated by the following symbols: †$p < .10$. *$p < .05$. ** $p < .01$. ***$p < .001$.
[f]Bold parameter estimates depict significant differences in model estimates between residents and specialists.

**Figure 1** Tested path model of job demands (workload, job insecurity, work-family conflict), job resources (autonomy, colleague support, supervisor support), personal resources (psychological capital, self-compassion, psychological flexibility), tenure, burnout and work engagement in a study on the role of personal resources for work engagement and burnout, 2017.[a–f]

### Job resources

Separate tests for both groups suggested that there were differences between residents and specialists regarding autonomy: autonomy was positively related to burnout for residents ($b$=0.19, p=0.016) but not for specialists (p>0.05). However, these differences did not significantly differ in a multigroup comparison test, p>0.05. Furthermore, separate tests for both groups suggested that colleague support differed between residents and specialists: it was not related to burnout for residents (p>0.05) but was negatively related to burnout for specialists ($b$=−0.41, p<0.001). However, these relationships did not significantly differ in a multigroup comparison test, p=0.088.

### Personal resources

Separate tests for both groups suggested that there were differences between residents and specialists regarding the personal resource psychological capital: psychological capital was not related to burnout for residents (p>0.05) but was negatively related to burnout for specialists ($b$=−0.58, p<0.001). A multigroup comparison

test confirmed that these relationships significantly differed, p=0.003. Furthermore, separate tests for both groups suggested that self-compassion differed between residents and specialists: self-compassion was negatively related to burnout for residents ($b$=−0.22, p=0.017) but not related to burnout for specialists (p>0.05). However, these relationships did not significantly differ in a multigroup comparison test, p=0.072. Also, separate tests suggested that there were differences between residents and specialists regarding the personal resource psychological flexibility: it was related to burnout for residents ($b$=−0.31, p<0.001) but not related to burnout for specialists (p>0.05). A multigroup comparison test confirmed that these relationships did significantly differ, p=0.003.

### Relationships with work engagement

The standardised path coefficients with work engagement as outcome are presented in figure 1.

### Job resources

Separate tests suggested that there were differences regarding colleague support between residents and

specialists: colleague support was not related to work engagement for residents (p>0.05) but was positively related to work engagement for specialists (*b*=0.49, p<0.001). A multigroup comparison test confirmed that these relationships did significantly differ, p=0.001.

### Personal resources

Separate tests for both groups suggested that there were no differences between residents and specialists regarding psychological capital: psychological capital was positively related to work engagement for both residents (*b*=0.37, p<0.001) and specialists (*b*=0.55, p<0.001). A multigroup comparison test confirmed that these relationships did not significantly differ, p>0.05. Furthermore, separate tests for both groups suggested that there were differences between residents and specialists regarding self-compassion: self-compassion was not related to work engagement for residents (p>0.05) but was negatively related to work engagement for specialists (*b*=−0.33, p=0.004). A multigroup comparison test confirmed that these relationships did significantly differ, p=0.003. Also, separate tests for both groups suggested that there were differences between residents and specialists regarding psychological flexibility: psychological flexibility was positively related to work engagement for residents (*b*=0.17, p=0.035) but not for specialists (p>0.05). However, a multigroup comparison test showed that these relationships did not significantly differ, p=0.778.

## DISCUSSION
### Main findings

The goal of this study was to gain insight on the prevailing demands and resources that contribute to burnout and work engagement in medical residents and specialists. This study revealed that residents and specialists face different demands in their work that cannot be measured by the same yardstick but instead require tailored solutions. Confirming prior studies on the stressful demands during residency,[7] our data showed that residents compared with specialists experienced less autonomy and felt more uncertain about the future of their job. Furthermore, symptoms of exhaustion and cynicism among residents increased with tenure, which may at least partly relate to growing feelings of job insecurity (see table 2). Contrary to what has been reported in previous studies,[50] residents did not report significantly higher exhaustion and cynicism than specialists but, on average, felt less engaged with their work.

We suggest that specialists and residents resort to different resources to cope with their job demands. Although both groups have the same level of resources at their disposal, only certain resources contribute to the well-being of specialists and residents, respectively. While specialists benefit from psychological capital and colleague support, residents benefit especially from psychological flexibility and self-compassion. It is likely that residents and specialists—as a function of their role

and the career phase they are in—use those resources that bring the greatest benefit when facing job demands at work. This will be discussed subsequently. In addition, psychological capital was found to play a role for the work engagement of both specialists and residents, which corroborates earlier findings among other professional groups.[51]

Physicians are exposed to high job demands, both during attendance and residency, which could harm their well-being.[12] However, our study suggests that job demands other than workload (eg, job insecurity, work-family conflict) and a lack of resources (eg, self-compassion, psychological capital and psychological flexibility) play a prominent role in the onset of burnout. The fact that the residents in this study reported a relatively lower workload than specialists, yet reported similar symptoms of burnout, underlines this notion. Generally, our findings suggest that for preventing burnout it is important to focus on those demands and resources that are most relevant for specific groups of physicians (eg, specialists or residents). Specialists may particularly benefit from interventions that raise their psychological capital and—at the team level—foster team cohesion and support, whereas residents may benefit relatively more from interventions that increase their self-compassion and flexibility.

### Resources that buffer burnout among specialists and residents
#### Personal resources

Consistent with prior research,[52 53] psychological capital played an eminent role for the well-being of specialists. Psychological capital may reduce the risk of burnout in two ways. First, it can counteract the distress associated with a demanding workplace by regulating negative emotions.[54] Second, individuals with high psychological capital tend to perceive job demands as challenges rather than hindrances. That is, they associate job demands with personal gain or growth,[55] evoking positive emotions, instead of fear and threat, evoking negative emotions.[55 56] Given the buffering capacity of psychological capital,[52 53] it is surprising that we did not find the same result among residents. Instead, we found that flexibility and self-compassion rather than psychological capital contributed to the well-being of residents.

The importance of flexibility and self-compassion among residents may be due to their specific career phase, which is characterised by insecurity, constant feedback and criticism. That is, residency is an extremely challenging period, in which residents have to deal with their newly gained responsibilities (ie, managing uncertainty, breaking bad news) while also processing new information and continuously adapting to new organisational structures and teams. Residents have to shift regularly between their roles as trainee and doctor,[57] experience high challenges at work and are confronted with their relative lack of knowledge and skills when entering residency. More so, residents are taught from medical school on to be critical towards themselves, a necessity that is

demanded in a high-stake work environment where care-lessness can have radical consequences. Consequently, medical professionals likely adopt a rather self-critical atti-tude.[58] One way to deal with these stressful work events is to accept one's inexperience, forgive one's deficiencies (ie, self-compassion) and remain effective despite self-doubt and worries (ie, psychological flexibility). More specifically, residents need to internalise that despite the current healthcare culture, not knowing, insecurities and mistakes are part of the journey and not a sign of weakness or failure. In addition, psychological flexibility may allow residents to shift between tasks and profes-sional roles, as it facilitates adapting to fluctuating situ-ational demands, shifting perspective and reconfiguring mental resources.[59] Thus, being kind towards oneself and viewing one's own shortcomings as human can help to safeguard residents against the stressors uniquely present in residency.

### Colleague support

In addition to psychological capital, colleague support also seems to promote the well-being of specialists. Numerous studies have indeed shown that social support is associated with both psychological and physical health outcomes[60 61] as it ameliorates the impact of stress and strain on health.[23 61 62] First, social support may involve emotional support, the feeling that one is loved and cared for.[63] Second, it may involve the provision and sharing of information.[63] Specialists work in relatively permanent teams with interdependent work relationships. Knowing their colleagues and their expertise well, they can ask for and receive emotional but also informational support. Colleagues can provide intimacy or reassurance during emotional and stressful events and they can assist in times of uncertainty and difficult medical inquiries that require another expert opinion.

Although our data does not allow any insights in the quality of support received from colleagues, it is possible that the quality of support is different for residents and specialists. Because of the nature of residency (eg, compe-tition and regular rotations), it is likely that colleague relationships are relatively less permanent and fruitful for residents. Although valuing their opinion, residents might not be convinced that they can ultimately lean on a fellow resident's opinion in solving medical problems. This could explain why residents benefit relatively less from colleague support than specialists.

### Resources that foster work engagement among specialists and residents
#### Personal resources

Our finding that psychological capital is a personal resource that is vital for the work engagement of both residents and specialists corroborates with prior studies among other professional groups.[54 55]

Unexpectedly, self-compassion was negatively rather than positively associated with the work engagement of specialists, while we found no such effect for residents. It is possible that high levels of self-compassion represent a self-protective bias, which serves to deny responsibility for failure.[64] That is, high self-compassion may lean towards attributing failure to external factors (eg, situational constrains, lack of help from others) allowing special-ists to maintain positive perceptions of their capabilities. Over a longer period of time, this way of thinking about their own shortcomings may hamper specialists' personal development and work efforts, which may ultimately cause a reduction in work engagement. It is therefore worth exploring whether the benefits of self-compassion depend on time or whether there is an optimal level—or perhaps a tipping point—at which self-compassion contributes to one's well-being.

### Colleague support

Colleague support not only buffered the occurrence of burnout but also fostered work engagement among specialists rather than residents. As argued above, special-ists as opposed to residents work in more permanent teams. It has been consistently found that the social support in these teams facilitates the work engagement of team members.[65]

### Study strengths and limitations

The participation of both residents and specialists from different specialties and the high response rate (78%) allowed for a realistic display of the demands and resources that physicians encounter during different stages of their career. Our research shows how different demands and resources relate to burnout and work engagement among specific groups of physicians and, as such, advances our understanding of how to intervene when well-being or work engagement are at risk. This is an important first step in the prevention of burnout and the conservation of work engagement among healthcare professionals. Yet, our results also indicate that burnout and work engagement are highly interweaved, just as in numerous prior studies.[65 66] Consequently, it is important to consider both burnout and work engagement when addressing job functioning as a whole.

Finally, our study has some limitations. First, although theory and research point to causal relationships between demands and burnout and resources and work engage-ment,[12 67] our design does not allow to draw causal conclu-sions. It is possible that the proposed relationships in the JD-R model are reversed. For instance, feelings of exhaus-tion and cynicism may change the way employees perceive their work demands, intensifying the feeling that demands are piling up.[68 69] Future research could use multiwave designs that can provide insight into the development of study variables over time and the causal dynamics in this process.[70] Second, all data have been gathered using self-report questionnaires. This might lead to a so-called 'common-method bias'.[71] A potential way to reduce this bias would be to expand the sources of information (eg, supervisors' assessments of employee burnout and engage-ment) and the methods of data collection (ie, triangulation

of data), for instance by including qualitative data as a next step. We are aware that self-ratings and observer ratings of work characteristics and job demands may not necessarily correlate high.[12] However, we believe that expanding the information source through third-party observations as well as triangulation can help to provide a richer picture of the work characteristics being studied.

Another limitation of this study is the relatively small sample size, especially in the group of specialists. To examine if power was sufficient in both samples of residents and specialists, we conducted post hoc power analyses (Soper DS. (2019). Post-hoc Statistical Power Calculator for Multiple Regression (Software). Available from http://www.danielsoper.com/statcalc) on several unsupported direct effects (eg, paths self-compassion and psychological flexibility to burnout for specialists, path psychological capital to burnout for residents). In all cases, the statistical power was 1.0, indicating that non-significant findings are most likely truly non-significant, that is, that this study had enough power for the conducted analyses. Furthermore, this study is limited by its sample composition, which predominantly consists of female pediatricians or pediatric residents. While the gender demographics of our sample closely matched the broader hospital population for residents, this was not the case for specialists as female specialists were over-represented in our sample. While this is likely due to the intervention context of this study, future studies should include a larger sample with different specialties and ensure a more equal gender distribution to test if these effects are stable across specialties and gender. However, with ample evidence supporting the JD-R model's presumptions in different professional contexts and in various samples, it is not likely that the results of this study are greatly biased by its sample characteristics. Finally, the concept of colleague support is limited, in the sense that it does not allow for a differentiation between support functions. Ideally, a concept of social support including such a differentiation[63] could help to disentangle how perceived social support helps specialists to counteract stress and exhaustion and promote work engagement.

## Implications

To our knowledge, this is the first study that attempts to reveal the specific demands and resources that may impact burnout and work engagement among residents and specialists. Understanding how demands and resources are linked to physician well-being and engagement is a fundamental premise for designing successful interventions to minimise the risk of burnout. Our results suggest that a one-size-fits-all approach might not be effective for promoting physician well-being but, instead, that interventions should be tailored to the specific needs of specialists and residents.[2] This is in line with a recent call to consider contextual complexities such as specialty or career stage when setting up interventions to promote physician well-being.[19] While interventions for specialists should focus on increasing psychological capital and colleague support, interventions

for residents should, in addition to increasing psychological capital, be aimed at increasing self-compassion and psychological flexibility. Interestingly, especially personal resources seemed to preserve physician well-being and engagement. Therefore, targeting personal resources rather than structural constraints seems promising to counter the demands physicians face. Additionally, interventions could also target training institutions and hospitals with the aim of building a culture that facilitates self-compassion, psychological capital and psychological flexibility among their residents and specialists. We consider testing the effectiveness of interventions aiming at fostering personal resources an important future inquiry.

## CONCLUSION

With physician well-being being central to optimal patient care, it is important to uncover work characteristics that influence work engagement and burnout. This study revealed that physicians are not a uniform body but that medical residents and specialists face different challenges in their work that require unique resources to resort to. While all physicians are likely to benefit from resources facilitating goal attainment (ie, psychological capital), medical residents may additionally benefit from self-care and flexibility and specialists may additionally benefit from social support. Finally, by respecting also the unique needs of residents and specialists, one can create equal opportunities for all physicians in the challenging workplace that healthcare is.

**Author affiliations**
[1]Work and Organizational Psychology, University of Amsterdam, Amsterdam, The Netherlands
[2]Pediatrics, Erasmus Medical Center—Sophia Children's Hospital, Rotterdam, The Netherlands
[3]School of Business and Economics, Vrije Universiteit Amsterdam, Amsterdam, The Netherlands
[4]Pediatric Hematology, Erasmus Medical Center—Sophia Children's Hospital, Rotterdam, The Netherlands
[5]Pediatrics, Leiden University Medical Center, Leiden, The Netherlands
[6]Pediatrics/Pediatric Intensive Care Unit, Erasmus Medical Centre—Sophia Children's Hospital, Rotterdam, The Netherlands

**Acknowledgements** The authors wish to thank all the residents and attending physicians who participated in this study.

**Collaborators** A M C van Rossum, W J W Kollen, R G M Bredius, A J Heesterman, M A van Houten, M J E Walenkamp, A A M Zandbergen, S C E Schuit, J E C Bromberg, A Willemse, S M van den Hee, M van den Heuvel and A Bakker-Pieper.

**Contributors** LS, AEMvV, TT, JK, APJdP and MdH contributed to the design and conception of the study. All authors were involved in the implementation of the study and the acquisition of the data. LS, AEMvV and JK analysed the data and interpreted the data together with TT, APJdP and MdH. All authors reviewed and approved the manuscript.

**Funding** The authors have not declared a specific grant for this research from any funding agency in the public, commercial or not-for-profit sectors.

**Competing interests** None declared.

**Patient consent for publication** Not required.

**Ethics approval** The institutional Ethic Review Board of the University of Amsterdam waived ethical approval for this study, on 12 December 2016; document 2016-WOP-7521.

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
