## [Reviewer comments · BMJ Open]

ARTICLE DETAILS

TITLE (PROVISIONAL)	Keep the fire burning: a survey study on the role of personal resources for work engagement and burnout in medical residents and specialists in the Netherlands
AUTHORS	Solms, Lara; van Vianen, Annelies; Theeboom, Tim; Koen, Jessie; de Pagter, Anne; De Hoog, Matthijs

VERSION 1 – REVIEW

REVIEWER	Tamara Robins The University of Adelaide, Australia
REVIEW RETURNED	06-May-2019

GENERAL COMMENTS	Introduction was well written overall. The topic is interesting and relevant. Method: Watch for small mistakes – i.e. page 9 line 26 ‘how often does your job require to work fast’ How many items does the Work Action and Acceptance questionnaire have? This is probably a word limit issue but some comment on validity of questionnaires and also directionality – i.e. higher score equal more flexibility, could be considered. I think it is important for the reader to understand more about the study that this was done in the context of especially with the reference to ‘signed up for coaching’. Usually a study like this would have a participants/procedure section. For this study it would be helpful to understand how the 278 were chosen – were all the doctors in the specialties invited or was the selection randomised? Were the participants representative of the make up of the doctors in the hospitals overall – i.e. how generalisable is the sample? Results are clear and presented well. Discussion page 22 line 17 – ‘resort to’ does this imply that they use different resources or that different resources are important in different areas? Another important limitation is power. There was no power analysis presented and a lot of statistical comparisons made. This should be addressed in the method and in the limitations section. Also, please discuss limitations of generalisability, particularly to the type of specialisations surveyed as well as addressing the earlier question of how similar the demographics of the included participants were to the overall hospital demographics. While personal resources appeared to be among the most important, organisational interventions could involve exploring the training institutions and hospitals and seeing whether the culture of the organisation facilitates self compassion, psychological compassion and psychological flexibility.
--

	Overall, a well considered study exploring the different impacts of demands and resources on residents and specialists working in the same environments. The findings were interesting and relevant and related in a useful way to future research and recommendations.
--	---

REVIEWER	Dr Yingchun Zeng The Third Affiliated Hospital of Guangzhou Medical University, Guangzhou, China
REVIEW RETURNED	15-Jun-2019

GENERAL COMMENTS	Thanks for offering this great opportunity to review this meaningful manuscript, titled as "Keep the fire burning: a survey study on the role of personal resources for work engagement and burnout in medical residents and specialists", this is a well written manuscript, but it needs to add information of sample size calculation, in the part of path analysis, it seems to be without enough sample size to conduct path analysis.
---

REVIEWER	Yen-Yuan Chen Graduate Institute of Medical Education & Bioethics National Taiwan University College of Medicine Taiwan
REVIEW RETURNED	30-Jun-2019

GENERAL COMMENTS	Thank you for this opportunity to review the manuscript. This study was pretty interesting, and made the readers to understand how job demands, situational resources, and personal resources relate to work engagement and burnout among specialists and residents. However, there are some insufficiencies which obscures strengths of the study. I tried to propose some suggestions about how this study could be further polished: Why was confirmatory factor analysis not conducted? For example, this study measured Workload using the four items of the Quantitative Workload Inventory. Are you sure the four items best representing the workload of the specific group in this study? To answer this question, to employ confirmatory factor analysis, for deleting items and/or collapsing dimensions (factors), is highly suggested. I would suggest to collapse Figure 1 and Table 4, to be a new Figure 1, composing all the information in a single figure.
--

VERSION 1 – AUTHOR RESPONSE

Reviewer 1

Introduction was well written overall. The topic is interesting and relevant.

We thank the reviewer for this comment, and are happy to hear that the topic of our paper is interesting and relevant.

Method:

Watch for small mistakes – i.e. page 9 line 26 'how often does your job require to work fast'

We thank the reviewer for this careful review. We have corrected this mistake (page 9, line 18): “An example item measuring quantitative workload is: “How often does your job require you to work fast?”

How many items does the Work Action and Acceptance questionnaire have?

We have added the number of items of the Work Action and Acceptance questionnaire (page 12, line 3): “Psychological flexibility, that is, the ability to flexibly take appropriate action towards achieving goals and values, even in the presence of challenging or unwanted events⁴¹ was measured with seven items of the Work Acceptance and Action Questionnaire⁴² ($\alpha = .81$)”

This is probably a word limit issue but some comment on validity of questionnaires and also directionality – i.e. higher score equal more flexibility, could be considered.

We agree with the reviewer’s comment. We revised the methods section that now contains internal consistencies in text (Cronbach’s alpha) for all scales (e.g., page 9, line 17, 25): “Workload was assessed with four items from the Quantitative Workload Inventory³² and two additional items ($\alpha = .85$).”; “The scale consisted of five items ($\alpha = .83$) including “Chances are, that in the future I won’t be able to find the job that I want” or “I am feeling insecure about the future of my career.” Changes appear on page 9, line 17, line 25, page 10, line 5, line 13, line 21, page 11, line 4, lines 10-11, line 20, and page 12, line 3, line 11, line 13, and line 19.

We also included a description of directionality (e.g., higher scores indicate higher frequency, i.e., higher workload) for all scales used (e.g., page 9, lines 21-22, page 10, line 3): “Higher scores indicate higher frequency, i.e., higher workload”, “Higher scores indicate stronger applicability, i.e., higher job insecurity.” Changes appear on page 9, lines 21-22, page 10, line 3, lines 9-10, line 19, page 11, lines 1-2, lines 5-6, lines 16-17, line 25, page 12, lines 6-7, lines 16-17, line 23.

I think it is important for the reader to understand more about the study that this was done in the context of especially with the reference to ‘signed up for coaching’. Usually a study like this would have a participants/procedure section. For this study it would be helpful to understand how the 278 were chosen – were all the doctors in the specialties invited or was the selection randomised? Were the participants representative of the make up of the doctors in the hospitals overall – i.e. how generalisable is the sample?

We agree with the reviewer and have added information on the context in which this study was conducted (page 8, lines 21-23): “Because this study is part of a larger program offering individual coaching to physicians, the sample consists of physicians that signed up for the coaching program or control participants.”

In the revised methods section we clarify how participants were chosen for participation in this study and that selection of participants was not random (page 8, lines 23-25, page 9, lines 1-2): The choice of departments and hospitals that were invited for participation in this study was based on internal logistics. That is, because the coaching program was only offered to physicians and residents from the pediatrics department, a relatively high number of participants in this sample are pediatricians.”

We also added information on the broader hospital population with regard to gender demographics in order to evaluate if our sample was representative of the make-up of doctors overall (page 9, lines 2-5): A comparison of the gender demographics of this sample with the broader population indicate that the sample of residents is representative of the hospital population. However, female specialists are overrepresented in our sample. Limitations of generalizability will be discussed.”

Results are clear and presented well.

We thank the reviewer for this comment. We are glad to hear that the results are clear and presented well.

Discussion page 22 line 17 – ‘resort to’ does this imply that they use different resources or that different resources are important in different areas?

We agree with the reviewer that ‘resort to’ could have different meanings and therefore have clarified its meaning (page 23, lines 13-14, lines 16-19): “Although both groups have the same level of resources at their disposal, only certain resources contribute to the well-being of specialists and residents, respectively.”, “It is likely that residents and specialists - as a function of their role and the career phase they are in - use those resources that bring the greatest benefit when facing job demands at work. This will be discussed subsequently.”

We believe this explanation illustrates that residents and specialists – although having the same level of resources at their disposal – use those resources that benefit them most in their specific work context.

Another important limitation is power. There was no power analysis presented and a lot of statistical comparisons made. This should be addressed in the method and in the limitations section.

We agree with the reviewer that we should discuss power issues. As a response we examined if power was sufficient in our sample of residents and specialists by conducting post-hoc power analyses for unsupported direct effects. We believe that these analyses confirm that power was sufficient for the path analyses we conducted (page 28, lines 4-10): “Another limitation of this study is the relatively small sample size, especially in the group of specialists. To examine if power was sufficient in both samples of residents and specialists, we conducted post-hoc power analyses¹ on several unsupported direct effects (e.g., paths self-compassion and psychological flexibility to burnout for specialists, path psychological capital to burnout for residents). In all cases, the statistical power was 1.0, indicating that non-significant findings are most likely truly non-significant, i.e., that this study had enough power for the conducted analyses.”

Also, please discuss limitations of generalisability, particularly to the type of specialisations surveyed as well as addressing the earlier question of how similar the demographics of the included participants were to the overall hospital demographics.

We thank the reviewer for this valuable comment. We have added information on generalizability of our sample in the methods section and also discuss this point in the section ‘study strengths and limitations’ of the discussion section (page 28, line 10-19). We agree that our sample is limited by its composition which predominantly consists of female pediatricians. However, we show that the gender demographics of our sample closely match the broader hospital population for residents. In order to generalize these results across specialties and gender, it is important that future studies include a larger sample with different specialties and ensure a more equal gender distribution (page 28, line 10-19): “Furthermore, this study is limited by its sample composition, which predominantly consists of female pediatricians or pediatric residents. While the gender demographics of our sample closely matched the broader hospital population for residents, this was not the case for specialists as female specialists were overrepresented in our sample. While this is likely due to the intervention context of this study, future studies should include a larger sample with different specialties and ensure a more equal gender distribution to test if these effects are stable across specialties and gender. However, with ample evidence supporting the JD-R model’s presumptions in different professional contexts and

¹ Soper DS. (2019). Post-hoc Statistical Power Calculator for Multiple Regression [Software]. Available from <http://www.danielsoper.com/statcalc>

in various samples, it is not likely that the results of this study are greatly biased by its sample characteristics.”

While personal resources appeared to be among the most important, organisational interventions could involve exploring the training institutions and hospitals and seeing whether the culture of the organisation facilitates self compassion, psychological compassion and psychological flexibility.

We agree with the reviewer that involvement of organizational interventions could be an important part of the solution when fostering self-compassion, psychological capital, and psychological flexibility. We have added this notion to the discussion section (page 29, lines 15-18): “Additionally, interventions could also target training institutions and hospitals with the aim of building a culture that facilitates self-compassion, psychological capital, and psychological flexibility among their residents and specialists.”

Overall, a well considered study exploring the different impacts of demands and resources on residents and specialists working in the same environments. The findings were interesting and relevant and related in a useful way to future research and recommendations.

We are very happy to hear that the reviewer appreciates our study and evaluates the findings as interesting and relevant.

Reviewer 2

Thanks for offering this great opportunity to review this meaningful manuscript, titled as "Keep the fire burning: a survey study on the role of personal resources for work engagement and burnout in medical residents and specialists", this is a well written manuscript, but it needs to add information of sample size calculation, in the part of path analysis, it seems to be without enough sample size to conduct path analysis.

We thank the reviewer for taking the time to make suggestions on our manuscript and are happy to hear that the reviewer evaluates the manuscript as meaningful and well written.

Just as Reviewer 1, Reviewer 2 comments on sample size calculation and power analysis. We agree with the reviewers on this point and have conducted post-hoc power analyses to examine if power was sufficient in our sample of residents and specialists. We conducted post-hoc power analyses for various unsupported direct effects and constantly found that power was optimal. Therefore we believe that these analyses confirm that power was sufficient for the path analyses we conducted (page 28, lines 4-10):“Another limitation of this study is the relatively small sample size, especially in the group of specialists. To examine if power was sufficient in both samples of residents and specialists, we conducted post-hoc power analyses² on several unsupported direct effects (e.g., paths self-compassion and psychological flexibility to burnout for specialists, path psychological capital to burnout for residents). In all cases, the statistical power was 1.0, indicating that non-significant findings are most likely truly non-significant, i.e., that this study had enough power for the conducted analyses.”

Reviewer 3

Thank you for this opportunity to review the manuscript. This study was pretty interesting, and made the readers to understand how job demands, situational resources, and personal resources relate to work engagement and burnout among specialists and residents. However, there are some

² Soper DS. (2019). Post-hoc Statistical Power Calculator for Multiple Regression [Software]. Available from <http://www.danielsoper.com/statcalc>

insufficiencies which obscures strengths of the study. I tried to propose some suggestions about how this study could be further polished:

We appreciate the reviewer’s time for comments on our manuscript and thank the reviewer for evaluating this study as interesting.

Why was confirmatory factor analysis not conducted? For example, this study measured Workload using the four items of the Quantitative Workload Inventory. Are you sure the four items best representing the workload of the specific group in this study? To answer this question, to employ confirmatory factor analysis, for deleting items and/or collapsing dimensions (factors), is highly suggested.

We agree with the reviewer that information on the factor structure of our scales would give meaningful insight in the quality of the data presented. In order to test if the items loaded on their respective scales, we conducted a number of confirmatory factor analyses. Additionally, because the predictors job demands, job resources, and personal resources each consist of three scales, we compared a three-factor model with a common factor model (page 13, lines 2-9, page 17, lines 1-17): “Factor structure. To examine whether the items loaded on their respective scales, we performed separate confirmatory factor analyses (CFAs) for the scales representing job demands, job resources, and personal resources, respectively. As each of these predictors consists of three scales, we compared a three-factor model to a one-factor model. We report the factor loadings and the commonly used model fit criteria, that is, the chi-square goodness-of-fit value, the chi-square divided by the degrees of freedom (CMIN/DF), the comparative fit index (CFI), the root mean square error of approximation (RMSEA) and the standardized root mean squared residual (SRMR).”, “Factor structure

All items loaded on their respective scales. Factor loadings were on average .73, .81, and .60 for job demands, job resources, and personal resources, with three items loading below .40 and a minimal loading of .28. These items were included because the scales were validated test instruments with overall high internal consistencies. The modification indices provided by the CFAs indicated that some items shared error variance. In order to improve the model fit, we allowed covariation of error variance between these items. Covariation was only allowed for items originating from the same scale. All three models provided adequate fit to the data with $\chi^2(86) = 153.74, p < .001, \chi^2/df = 1.79, CFI = .95, RMSEA = .06, SRMR = .06$; $\chi^2(180) = 312.75, p < .001, \chi^2/df = 1.74, CFI = .97, RMSEA = .06, SRMR = .06$, and $\chi^2(262) = 435.96, p < .001, \chi^2/df = 1.66, CFI = .90, RMSEA = .06, SRMR = .07$ for the three-factor models representing job demands, job resources and personal resources, respectively. Our results showed that the hypothesized three-factor model of these predictors provided a better fit to the data than a common factor model (e.g., fit indices of the common factor model of job demands: $\chi^2(89) = 864.96, p < .001, \chi^2/df = 9.72, CFI = .46, RMSEA = .21, SRMR = .19$). The differences in the chi-square goodness-of-fit value between the three factor and the common factor models were significant, all p ’s $< .001$.”

You can find the fit indices and model comparisons in the table below. In short, results showed that the three-factor model provided a significantly better fit than the common-factor model for job demands, job resources and personal resources.

		χ^2	df	χ^2/df	p	CFI	RMS EA	SRMR	model comparison	$\Delta\chi^2$	Δdf	p
	Job demands											

1	common factor model	864.96	89	9.72	.00	0.46	0.21	0.19				
2	3-factor model	153.74	86	1.79	.00	0.95	0.06	0.07	model 2 > model 1	711.22	3	<.001
	Job resources	χ^2	df	χ^2/df	p	CFI	RMS EA	SRMR	model comparison	$\Delta\chi^2$	Δdf	p
1	common factor model	1819.70	183	9.94	.00	0.58	0.22	0.24				
2	3-factor model	312.75	180	1.74	.00	0.97	0.06	0.06	model 2 > model 1	1506.95	3	<.001
	Personal resources	χ^2	df	χ^2/df	p	CFI	RMS EA	SRMR	model comparison	$\Delta\chi^2$	Δdf	p
1	common factor model	760.60	265	2.87	.00	0.72	0.10	0.10				
2	3-factor model	435.96	262	1.66	.00	0.90	0.06	0.07	model 2 > model 1	324.64	3	<.001

I would suggest to collapse Figure 1 and Table 4, to be a new Figure 1, composing all the information in a single figure.

We have created a new Figure 1 that now combines information that was earlier presented in Figure 1 (path model) and Table 4 (standardized path coefficients). We agree with the reviewer that this information is now presented more succinctly (see page 39 for figure legend, and see apart attachment for Figure 1).

VERSION 2 – REVIEW

REVIEWER	Dr Tamara G. Robins The University of Adelaide, Australia
REVIEW RETURNED	21-Aug-2019

GENERAL COMMENTS	The revision suggestions were addressed well and thoroughly! The article reads very well and adds relevant information in an area of current concern.
--

REVIEWER	Dr Yingchun Zeng The Third Affiliated Hospital of Guangzhou Medical University
REVIEW RETURNED	02-Aug-2019

GENERAL COMMENTS	no more additional comments.
------------------------------

REVIEWER	Yen-Yuan Chen National Taiwan University College of Medicine Taiwan
REVIEW RETURNED	21-Aug-2019
GENERAL COMMENTS	I have no further comments.